

# Early engagement with First Nations in British Columbia, Canada: A case study for assessing the feasibility of geological carbon storage

Katrin Steinthorsdottir[1], Shandin Pete[1], Gregory M. Dipple[1], Richard Truman[2], Sandra Ósk Snæbjörnsdóttir[3]

[1]Department of Earth, Ocean and Atmospheric Sciences, The University of British Columbia, Vancouver, BC V6T 1Z4, Canada
[2]Geoscience BC, Vancouver, V6C 2T7, Canada
[3]Carbfix hf., Reykjavík 110, Iceland

*Correspondence to*: Katrin Steinthorsdottir (ksteinth@eoas.ubc.ca)

**Abstract.** This work describes early engagement with 21 First Nations or alliances, that represent 41 Nations, in British Columbia, Canada. Geological researchers conducted this work as a case study to assess the feasibility of carbon storage in serpentinite rocks. The priorities for engagement were to inform people about the project and its implications, get consent for fieldwork, have a discussion, and start building relationships. Aside from the geology and logistics of a site for a carbon storage project, the permitting and acceptance by the local community and the traditional lands' rightsholders are needed for a successful project.

The engagement levels and timelines varied from short phone calls to emails and video meetings. The general reception was positive, and people showed an interest and appreciated being contacted early. Common areas of discussion were water quality, salmon habitat, and involving the youth. This work outlines the first step for engagement, and further work will be done if a proposed $CO_2$ storage project is to proceed.

## 1 Introduction

One of the initiatives that is needed to reach global climate goals is to capture $CO_2$ and store it safely and permanently (IPCC, 2021; Fuhrman et al., 2024). This can be conducted in mafic formations, such as basaltic rocks, as has been demonstrated using the Carbfix technology (e.g. Snæbjörnsdóttir et al., 2020), in ultramafic formations, such as serpentinite (e.g. Goff and Lackner, 1998; Kelemen et al., 2011), or in sedimentary rocks via conventional storage methods (e.g. Furre et al., 2017). Furthermore, carbon dioxide removal (CDR) and carbon capture and storage (CCS) must be initiated and operated in a just way for local communities (e.g. Bushman and Merchant, 2023; Goldberg et sal., 2023). Community acceptance can often be increased by good communication practices and knowledge transfer (Desbarats et al., 2010; Wallquist et al., 2010; Brunsting et al., 2011a; Wallquist et al., 2012; Haug and Stigson, 2016).

Research and practice have shown that to get a successful carbon storage project up and running, it is critical to have acceptance, support, and partnerships with local communities (Carbon Business Council, 2023; Satterfield et al., 2023).



Examples of when engagement at early stages was not done successfully or concerns came up which led to a stop or a halt in a project are the ocean alkalinity project in Cornwall, England (Weeks, 2023), the $CO_2$ injections in Barendrecht, Netherlands (Brunsting et al., 2011b), and In Salah, Algeria (Verdon et al., 2015; Carbon Capture and Sequestration Technologies program at MIT, 2016).

Many areas globally that host volumetrically large mafic or ultramafic formations where a carbon storage project via mineralization could be done are on Indigenous lands, such as in interior British Columbia (B.C.), Canada (e.g. Mitchinson et al., 2020). Currently, no large-scale carbon storage project has been implemented in B.C. but it is being explored (e.g. Geoscience BC, 2023; CICE, 2024; Solid Carbon, 2024), although acid gas ($H_2S$ and $CO_2$) have been injected for disposal in sedimentary rocks (Bachu and Gunter, 2005) and forest carbon emission offsets have been generated (Coastal First Nations,

2022; Connolly, 2022). For a carbon storage project, Indigenous Peoples will both be the ones affected and have the opportunity to benefit, such as is also the case in renewable energy or mining projects (Schlosberg and Collins, 2014; Dharapak, 2022; Parmenter et al., 2023; Jones, 2024). Both B.C. and Canada have now endorsed the United Nations Declaration on the Rights of Indigenous Peoples (UNDRIP) which includes free, prior and informed consent for projects (British Columbia, 2019; Government of Canada, 2024b). As has been discussed extensively, for example at the First

Nations Major Project Coalition conferences (FNMPC Conference, 2023), shared decision-making and equity partnerships are the way forward and the first step for that is engagement (Wilson-Raybould, 2022).

This study presents a case of early engagement with First Nations in B.C. for fieldwork and a project concept of carbon storage in serpentinite rocks (e.g. Geoscience BC, 2024). The priorities for engagement were to inform people about the project and its implications, get consent for fieldwork, have a discussion, start building relationships, and build

understanding before any development is proposed. Early engagement can be challenging to navigate, takes time, needs self-reflection, and is vital to starting a project (e.g. Haggart et al., 2011; Smith and McPhie, 2022).

## 2 Background of project

The Carbfix $CO_2$ storage technique has been proven in basaltic rocks in Hellisheidi, Iceland (e.g. Matter et al., 2009; Matter et al., 2016; Clark et al., 2020). The $CO_2$ can either be captured from point source emissions or captured directly from

the atmosphere. The $CO_2$ is then dissolved in water and injected into reactive and porous rock formations, forming safe and permanent carbonate minerals (e.g. Snæbjörnsdóttir et al., 2020). One of the reasons to study if serpentinite can work for this method is to open up new areas around the world not previously considered for geological $CO_2$ storage, which increases the opportunities to pair sinks and sources, and effectively decreases transport costs.

The project concept is to find a place in B.C. to research the feasibility of serpentinite for $CO_2$ storage via shallow

injection. This project is a collaboration between researchers at the University of British Columbia (UBC), Geoscience BC and Carbfix. The location primarily depends on 1) the geology, a volumetrically large serpentinite is present; 2) the logistics, such as electricity, a water source and access; and 3) permitting and acceptance by local community and traditional land





owners and rightsholders for fieldwork, drilling and a pilot injection. Work to date has revealed many areas in B.C. that show potential (Mitchinson et al., 2020; Cutts et al., 2021). Three of those areas were chosen to do further work on, start

early engagement and collect geological data.

We engaged with multiple First Nations in B.C. with traditional lands encompassing the three chosen sites. All of the sites are in areas where First Nations do not have treaties in place with the government, and where they have not ceded sovereign rights as Nations. These First Nations and alliances representing multiple Nations, vary in size and history, from ca. 50 to 3000 people each. The objectives of the early engagement for the project were to: provide information about the

project and $CO_2$ storage potential; ask for consent to access the land and conduct fieldwork; ask if they have any concerns, criteria or recommendations for fieldwork, such as being accompanied by a representative; ask if we should reach out to anyone else; and propose collaboration or ask if they have any suggestions or recommendations for this project or potential future stages of research.

## 3 Methods

Before any community engagement was started, an engagement and adaptation process plan was set up (e.g. Gamble and McQueen, 2019; Association for Mineral Exploration, 2020; Office of Indigenous Strategic Initiatives, 2020; Kennedy and Keenan, 2023; Coastal Conservancy, 2024), summarized in Fig. 1. This work was started nine months (November 2022) before anticipated fieldwork. Since there is no systematic engagement system in B.C. (e.g. Government of Northwest Territories, 2024), it can be hard to navigate the best way for such outreach. However, some Nations have their own referral

system and when that was the case, we followed that process. The engagement plan included researching potential implications of the planned work and research, especially for local communities, Indigenous Peoples, and the ecosystem, and the socioeconomic state and historical or recent work in the areas. Additionally, researching respectful engagement practices (e.g. Adams et al., 2014; Wong et al., 2020; Smith and McPhie, 2022; Reid et al., 2024) that included learning and reflecting on how we might have a different way of knowing or worldview than many communities (Wilson-Raybould, 2022;

McGregor et al., 2023). An individual's worldview can be shaped by their culture and education and impacts presuppositions, beliefs and actions (e.g. De Santo et al., 2023; Oxford Reference, 2024).

While working on this engagement process, we submitted an ethics application to the Behavioural Research Ethics Board within UBC (H23-02376). The Behavioural Research Ethics Board reviews research by and with Indigenous Peoples and communities including research on Indigenous lands and traditional place-based knowledge (UBC Office of Research

Ethics, 2024). As most geoscience work does not consider ethics approval, there was uncertainty around the requirements for when an application should be submitted. We wanted to characterize both the constraints around the dissemination of the engagement process itself and the learnings on the way from the Indigenous representatives we discussed with. It was decided and agreed upon that for this topic, we did not need to complete the ethics review process as long as we kept Nations and representatives anonymous and did not document others' knowledge.



95 We compiled the names of communities whose land and territory encompass the field sites (e.g. First Peoples' Cultural Council, 2021; British Columbia, 2023; BC Assembly of First Nations, 2024; Native Land Digital, 2024). There are 45 First Nations and First Nation groups within the field areas. We started reaching out seven months (January 2022) before the anticipated fieldwork, beginning with the First Nations alliances. When a phone number was available, a call was made to the general offices of First Nations/alliances listed within consultation areas. When the contact was by phone, either the first

100 time calling or later, there was an introduction to the project and an inquiry on whom we should talk to regarding prospective fieldwork. In most cases, we were referred (via phone or email) to someone, such as a natural resource director or lands manager. When that was not the case, voice messages and/or emails to general email addresses were left.

 We had notes ready to guide the discussion for the phone calls, and for the voice mail, a short (<60 sec) written out introduction. For following up after phone calls or through primary emailing, we had ready a one-page description to send,

105 listing the project background, prospective fieldwork, possible implications, and an offer to meet via phone or video call to discuss (Supplementary Data A). In most cases, there was a follow-up again a few weeks later and again at a later time.

 For some Nations, there were further discussions with representative/s via a phone call or video meeting. We had a few slides ready with background, fieldwork objectives, timelines, maps, and implications in plain English. Topics of discussions varied in content and detail. It often included if this is an area of priority for the community, collaboration, meeting to

110 discuss further before fieldwork, sample collection and data storage, other suggestions and criteria. The timing of reaching out and conversations, who talked to, phone numbers and email addresses, and what was discussed were documented. Furthermore, a plan for post-fieldwork follow-up and dissemination of any work done, results, outcomes, and possibly future collaboration.



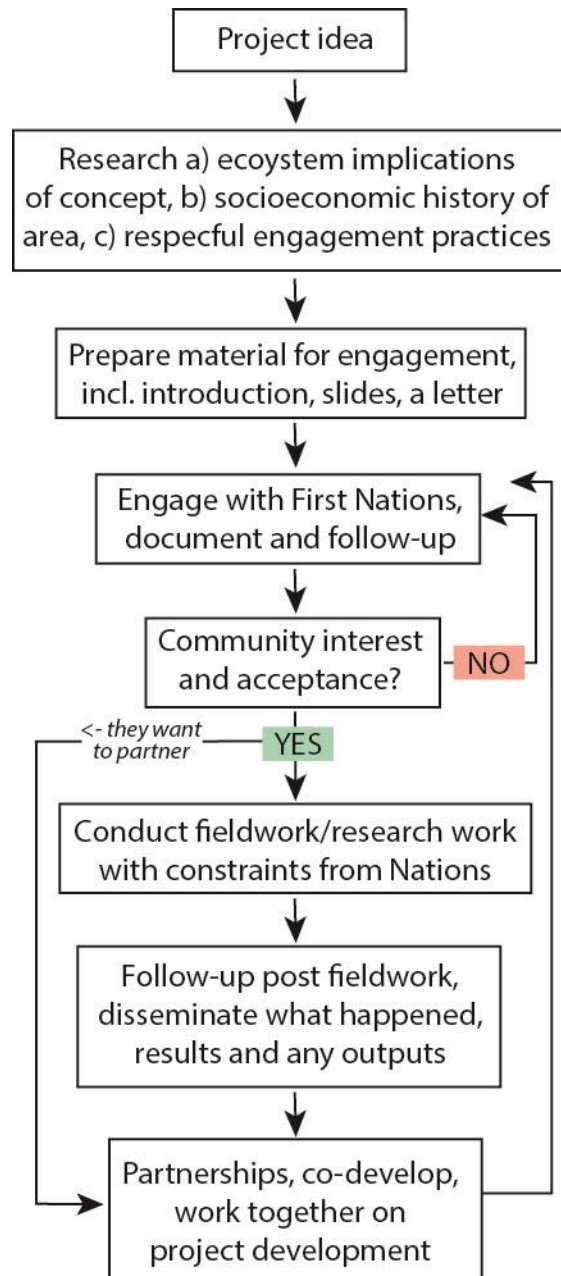

**Figure 1. Overview of engagement plan for the project.**


## 4 Outcomes

In total, there are 45 First Nations and First Nation alliances (either tribal councils, Nation alliances or joint ventures) represented by 25 Nations or alliances. The engagement levels that were reached are tabulated in Fig. 2. We contacted the





seven alliances first. Out of those alliances, one told us to contact the First Nations, which they represent directly, and

another was no longer functioning. Three Nations did not respond. For the rest of the 21 Nations or alliances, that represent 41 Nations, we reached some phone or email conversations about fieldwork for 14 of them. Three answered that the areas were not within their traditional lands; for five, there were back and forth phone or email conversations; for six there were video meetings.

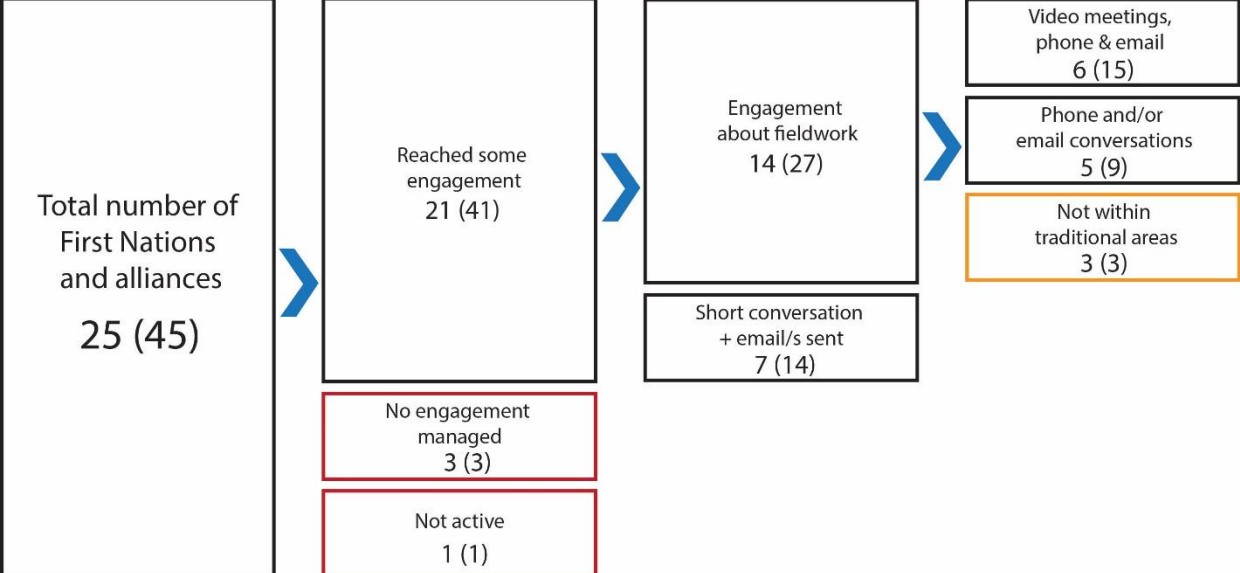

**Figure 2. Engagement levels and number of First Nations and alliances. The first numbers represent the number of Nations or alliances that represent the number within parentheses (e.g., there were 25 Nations or alliances that represent 45 Nations).**

After initial contact, through a phone call or email, we sent the general letter the same day and followed up via email two to three weeks later. In some cases, if we heard nothing back, we followed up again a few weeks later. Two Nations answered the first email after a phone call without a follow-up. Two other Nations replied that they would like to meet but

did not reply again about when. For the representatives that we met for a video meeting, it happened between 3.5 to 5.5 weeks from the first phone call. In Fig. 3, total hours have been accumulated for the engagement process for one Nation (~43-49 hours) and the project in total, representing roughly 124-264 hours or 15 to 33 days. Additionally, there is an approximate timeline shown for the engagement process from the start of getting an engagement plan ready in November 2022 till follow-up conversations in May to November 2023.





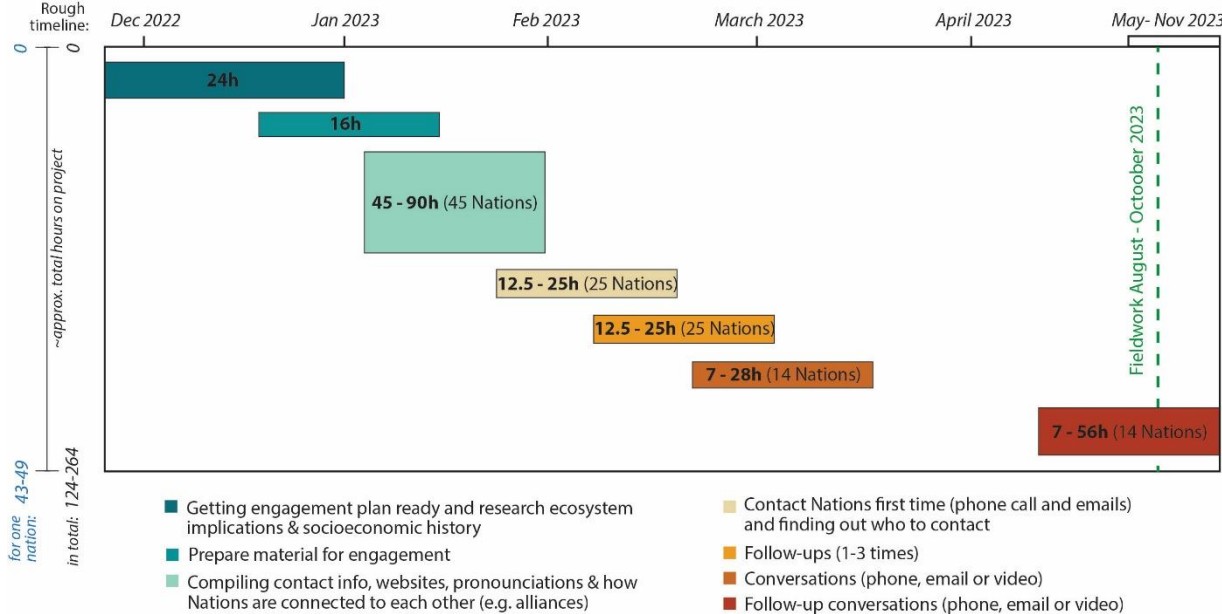

**Figure 3. Rough timelines and approximately total hours for the engagement process for one Nation and in total.**

Three of the Nations pointed us toward other Nations closer to the site. Some Nation representatives asked for more information regarding the location, on what the field plan encompassed, and when and who would to be there. The general reception was positive for all the conversations that took place through a phone call or video meeting. Many representatives showed an interest in the project, knowing more about opportunities, future collaboration, and how to get the youth involved. A few people mentioned an appreciation for being contacted so early. One discussion included the idea of having the project put into their newsletter. Some people expressed surprise that the project was different than mineral exploration or mining. Common themes were around water quality and salmon habitat.

Two Nations showed an interest in having a representative from the Nation join fieldwork. One because of possible later collaboration and to assist. However, this did not work out due to timing schedules. The other Nation sent a cultural monitor to join fieldwork to monitor and assist in trail finding and other logistics. Because of wildfires, the timing of fieldwork got changed and delayed, but the Nation reached out at the end of the summer to check in and see if we were still interested in coming. During planning and in the field, there were multiple discussions with the cultural monitor and others about the field area. This included sharing of Indigenous Knowledge, such as place-based stories, land use, animal habits, and stream, trail and outcrop locations. We will not report on these data as they are not our findings, we do not have permission from the people, to be respectful, and due to constraints around the ethics application from the university. The fieldwork was carried out successfully except for in one instance. Although we had support from the First Nation, a local individual did not support our work and so we halted our field studies.



## 5 Discussion

### 5.1 Engagement levels

The engagement for the 25 First Nations or alliances went generally well. However, the depth of engagement, discussions, and timelines varied between Nations, from no responses to short phone calls to multiple meaningful video meetings (Fig. 2 and 3). This variability can have many reasons. It is recognized that First Nations, in some cases, do not have sufficient resources or staff to engage in all requests. Lack of responses might also reflect external impacts such as

wildfire, insufficient interest in the project, or no sense of relevance. The engagement levels likely also reflect each Nation's different sizes and capacities.

       Engagement with 14 Nations or alliances reached a discussion about fieldwork (Fig. 2). With all of these conversations, the representatives showed an interest in learning more. Some representatives were excited about the benefits of storing $CO_2$ and the opportunities that could come with it for their Nation, especially their youth. Surprisingly, there were no negative

reactions about $CO_2$ storage, but rather questions about the implications for the water quality and fish health. Some asked if we would use a helicopter for the fieldwork, if drilling were involved, and where we stayed and cooked. In some instances, these questions were related to possible disturbances for wildlife or potential partnerships with their Indigenous businesses. There was also an appreciation of being contacted so early, months before anticipated fieldwork, especially since we were only doing simple fieldwork, not a multiple-person team and drilling.

Coming from a technical background, we were expecting questions about how this project might impact seismicity and if it is similar to fracking, and thus, we showed and discussed those differences. There was not much conversation about this but rather about how this project was mining-related. A few people were surprised, in a somewhat positive way, to learn that the project was not mining related. This could be because of the local geology that hosts mineral deposits, past experiences with mineral exploration companies in the region, and that oil and gas regions within B.C. are far away. Positive engagement

at this time might also be due to the project being at a research stage, working with a not-for-profit, the purpose of it is climate action, and being open to fieldwork changes, questions, and collaboration. There was more interest and support from the Nations than we expected.

### 5.2 Representatives and relationships

       We talked to and met administrative assistants, referral coordinators, land managers, natural resource directors, and

chiefs through the engagement process. These people have different roles and authorities within their Nations or alliances. During the initial outreach, they were often unsure who to forward our request to, as this kind of outreach is seemingly rare. In some instances, we were referred to housing or education departments, which then referred us to other departments, such as lands or resources.

       Similarly, which Nation or alliance took "ownership" of the project varied. Some of the alliances of Nations that we

talked with were the spokespersons for those Nations they represent. In one instance, they brought in a representative of one



of the Nations to a video meeting, and in other cases, they sent the contact information of the one or few Nations that should be contacted. In another case, the alliance's representative told us to contact each Nation separately. This shows how complex it can be and the amount of work and time which needs to be allocated for understanding when it's respectful to contact a Nation directly that is part of some alliance.

In some instances, when a conversation was reached, a Nation's representative told us to contact another Nation but would like to be kept in the loop during the project development. In another instance, a representative inquired if we would have a monitor from another Nation that was closer to the area join fieldwork; then, there would be no need for them to send a monitor. This is different than on a previous project, and fieldwork some of the authors took part in, where two Nations have traditional lands in the area, and they both sent a monitor for the fieldwork (Steinthorsdottir et al., 2020). Additionally,
it varied if the representative we interacted with asked if other Nations in the area had been contacted or not. This shows how varied the relationships are between Nations due to, e.g. their different opinions, their history working with each other, or other reasons.

    There may be different reasons why the Nations that showed an interest in sending a monitor or representative to join the fieldwork chose to do so. One of the Nations has a consulting service and a geologist who works there who wanted to
join fieldwork to learn about the carbon storage, what that entailed in the field, and the possibility of collaborating on a future project. For the other Nation that sent a cultural monitor, it seems to be their criteria for either doing research or industry work on the land, possibly partly because they have the resources and want to ensure that work is respectfully conducted. They showed interest in the project and discussed the possibility of collaboration.

    There are many advantages of having a representative or someone from the area accompanying fieldwork, and it should
be recommended or even enforced on Indigenous lands, for research, government or industry purposes (e.g. Association for Mineral Exploration, 2020). However, this, of course, depends on each Nation and their willingness and resources. At the very least, approval of planned work and a discussion about the area is helpful before engaging in fieldwork. A lot of information is hard to find without local input, even with the help of Google Earth, blog posts, or news articles, such as some trails, road conditions, landslide damages, or recent encounters with wild animals. For our fieldwork, it benefitted us to have
a representative joining, especially when we met a Nation's member in the field who did not appreciate us being there unannounced. It immensely helped us that the representative was accompanying us and that we had engaged with and discussed with representatives from the other Nations whose traditional lands also encompasses the area. Additionally, a significant benefit for both the researchers and the Nations of successful engagement is having positive relationships for any future development and collaborations for a project.

As has been shown by recent events, such as at the Juukan Gorge in Australia (Antar, 2023) and by the San Andres mine in Honduras (Radwin, 2022), people, in some cases geoscientists, have different end-goals or knowledge and that can affect what and where they mine or sample. This can lead to negative impacts on spiritual or archaeological sites. Furthermore, understanding different opinions and perspectives of rightsholders and stakeholders can be critical for project development and decision-making. Different opinions can affect trivial things such as not being allowed to sample rocks, as happened to



us, not accessing roads or getting permits to drill. This might not have significant consequences at the time but can have meaningful implications for timelines and sunken costs, thus, for a project's success. Additionally, for the First Nations, there may be potential consequences in regard to Sovereign rights and further exacerbating historical trauma.

## 5.3 Geoscientist responsibility and role

Research has shown that to get a successful project up and running, community support or involvement from the
beginning of a project can help things go smoother and on shorter timelines (e.g. Wilson et al., 2016; Mathisen, 2021; FNMPC Conference, 2023; Jones, 2024). Community support can be achieved if the community owns and leads a project, such as the projects Tu Deh-Kah geothermal (Tu Deh-Kah, 2024) and Atlin's hydropower (THEL, 2024) in northern B.C. However, what is not often incorporated in timelines is the time frame to get the community support or partnership in the beginning, the early engagement. As we experienced, even finding out who to talk with takes weeks to months, and the
follow-ups and decision-making take even longer (Fig. 3). It may be well known, but we want to emphasize the importance of having the discussions and getting the information about a potential project and its implications as early as possible to a community and having that information in an easy-to-understand format (e.g. Mackenzie et al., 2020). Indigenous or local communities opposition can delay or stop a project (e.g. Lavoie, 2018; Centre for Social Responsibility in Mining, Sustainable Minerals Institute, 2023).

Geoscientists are often the first ones interacting, answering questions, and meeting the people living in the area before or during fieldwork. Scientists can also have an important role in getting innovative projects up and running (e.g. Becattini et al., 2024). Although they are trained in many of the technical considerations for a project, they are not trained in socioeconomic considerations. Some considerations to keep in mind are shown in Fig. 4 and range from the global scale of climate change effects to individual opinions (e.g. Snæbjörnsdóttir et al., 2020; Huggins et al., 2023). Different perspectives
and views might cause friction that could be improved with training in science communication, Indigenous socioeconomic history, and engagement practices. Typically, these aspects are not included in a geoscientist job description but this is changing as there is a shift happening in B.C. and elsewhere. More people are acknowledging that early engagement is essential and takes time. This can be seen in changes in the industry, government, and research practices (e.g. McGregor et al., 2016; Association for Mineral Exploration, 2020; Office of Indigenous Strategic Initiatives, 2020; Rogers et al., 2022a;
FNMPC Conference, 2023; Stein et al., 2024). In the earth sciences department at UBC, which some of the authors are affiliated with, we are implementing a guidelines document for engagement practices with Indigenous People that was in part developed from this case study. In some cases, there is a duty to consult, such as for federal employees (Government of Canada, 2024a) and there is the recent court decision to change the Mineral Tenure Act, the process of staking mine claims in B.C., to include Indigenous consultation (Abell, 2023). However, considerable work still needs to be done within
geoscience to change the narrative, take responsibility (Gillette, 1972; Peppoloni and Di Capua, 2017), and move away from colonization practices that geosciences are in many ways still linked with (Sangwan, 1994; Pico, 2019; Cartier, 2021; Gewin, 2021; Radwin, 2022). Below we discuss several topics to improve for the geosciences.





**Figure 4. Technical and socioeconomic considerations on different scales when it comes to implementing a successful carbon storage via mineralization project.**


### 5.3.1 Communication and collaboration

A communication strategy and communicating simply, yet in-depth, to a diverse audience, is critical, can be difficult, and often needs preparatory work and flexible timing. Furthermore, it is currently not traditionally encouraged or taught



within the geosciences (e.g. Wong et al., 2020; Rogers et al., 2022). If researchers collaborate successfully with Indigenous

Peoples, an essential part of that will be effective communication and dissemination before, during, and after the project starts. Even if no collaboration is planned, or there are not obvious implications of the project, researchers should still engage and discuss with the land's rightsholders.

For projects such as carbon storage, early engagement outlining risks and benefits and having the local community in the loop from the beginning can help with project development (Brunsting et al., 2011a). Furthermore, if the community has

all the information, they can make informed decisions and even participate in the project as project leaders or partners. The local community is likely to be most affected in case of gas leaks or infrastructure changes and reap the benefits of job opportunities and possibly project ownership (Bushman and Merchant, 2023; Goldberg et al., 2023; Low et al., 2024).

### 5.3.2 Meaningful engagement

To reach meaningful engagement, we first need to assess what that entails. It will look different for Nations or

communities, types of research, and development stages of projects (Wilson et al., 2016; Plunk and Gehlert, 2018). In general, meaningful engagement means a back-and-forth conversation, including listening and involvement, and that the communities' needs or priorities are met in some way. That could be everything from a long phone call to several meetings, the opportunity to join fieldwork, sign official documents, or collaborate (Adame, 2021). As shown in this study, early engagement can take months, and reaching meaningful engagement can take many years. This can be a conundrum for

academic researchers and graduate students as timelines of project scopes and funding are often relatively short, and the engagement timelines are not usually incorporated (e.g. Adams et al., 2014).

In many instances, it was not clear which department within the First Nation was an appropriate contact to host engagement because the nature of engagement was unfamiliar. Additionally, the researchers' lack of experience with engagement work made it, in some instances, confusing and there was uncertainty around protocols. It might be beneficial

for both sides to have the start of engagement more streamlined, with some engagement recommendations or system (e.g. Nunavut Research Institute, 2021; Government of Northwest Territories, 2024; Government of Yukon, 2024). A streamlined system could help with the uncertainty of whether and when engaging with communities is appropriate. Some fieldwork research can feel like it falls in the cracks due to them being small or short projects, such as collecting a rock sample, picking up float samples, gathering water samples or non-destructive analyses or observations. Additionally, it might encourage

engagement to start earlier, as we were told that it was a positive surprise that we reached out months ahead of time.

### 5.3.3 Reflecting on roles

To prepare for engagement, one needs to reflect on multiple considerations (e.g. Fig. 4), perspectives, and your and others' identities. Give yourself time to learn about communities' values, norms and culture and how to communicate and behave respectfully (e.g. Wong et al., 2020; Smith and McPhie, 2022). Your identity can have benefits and implications for

first impressions, connecting with people, and interpreting the project or findings. Researchers must take time to self-reflect



on their privilege before and during engagement and how it affects our worldview, the questions we ask, how we do research, and how we plan for and feel during fieldwork. This includes your ethnicity, gender, your educational background, but also who you, or the project, are affiliated with, such as a university, local community, or a company, and if there are any biases or stereotypes associated with them (e.g. Wilson-Raybould, 2022). Additionally, differences in specialties, language

skills, cultural background, and what words are used to introduce the project can affect first impressions or cause friction (Smith and McPhie, 2022). We realized through this process that it is better to talk about how the project is different compared to other projects that may have a bad reputation from the start (e.g. fracking or mining) instead of being asked later because there is more uncertainty around it.

The role of the representative/s you discuss and meet with during engagement can vary. Their past experiences with

other researchers or their relationships with the land might affect their professional or private opinions, impressions, and/or consent. Furthermore, opinions can be different between representatives and other members of the Nation, which can affect decision-making. For example, in one of our previous projects, where there was an active non-disclosure agreement with a mineral exploration company, we discussed and got consent from the Nation's chief to conduct fieldwork. We then had a follow-up meeting with the elders of the Nation, who had different questions and opinions. However, the meeting ended with

an agreement and a memorandum of understanding between the Nation and the researchers. During this project, there was the example of meeting a Nation member that did not allow sampling of rocks even though the representatives we discussed with had previously allowed it. This clearly shows how important it is to have good communication skills, reach out early, and engage meaningfully.

### 5.3.4 Data ownership

For this project, we had discussions with Nation representatives about data collection and sharing and what they expected. In the follow-up plan, there is the commitment to share both the preliminary results from fieldwork and the full results when ready. This discussion is relevant because to get to an equitable partnership, all involved need to have the data early on to make their own decisions, initiate, or be part of a project (Kennedy and Keenan, 2023). This is especially important for projects that may affect local communities, such as leakage, oil spills or mining potential.

The unceded, traditional and ancestral lands of many First Nations in B.C. are now on what is called "crown" land, that is, public land or waters that are "owned" by the provincial government. The discussion around who owns the land, who owns the right to it, and to develop it is a complicated matter that is being scrutinized (Simmons, 2022) and, in some cases, revised (e.g. Abell, 2023). For discussion purposes here, we focus on collecting, owning and disseminating data generated on traditional lands, such as geological fieldwork. That includes letting the Nations know and getting permission to collect data

(communication and engagement), even if that data does not have any apparent implications at the time for the communities (e.g. Nyblade and McDonald, 2021). Once raw and interpreted data are collected, there is uncertainty around who owns it, how and where it is stored and shared with other stakeholders or interested parties. Once the data has been written up, put into models, or compared to previous work, which often takes years, it needs to be defined where and to whom those





interpretations go (Nyblade and McDonald, 2021; FNIGC, 2024; GIDA, 2024; USGS, 2024). Sometimes, data is published

as part of theses or journal articles, often behind a paywall, or disappears in a notebook.

### 5.3.5 Responsibility for climate action

For geoscience, we study and work on many environmental and climate action topics to help society. These include assessing and mitigating natural hazards, finding and developing renewable energy sources (e.g. geothermal), mineral exploration for metals, monitoring contamination and groundwater resources, and storing $CO_2$. Additionally, we need to

acknowledge that geoscience is rooted in many ways in colonization practices and has a troubling history with resource extraction on Indigenous lands (e.g. Radwin, 2022). It might be the duty of today's geoscientists to use our knowledge of the earth systems to help with climate action projects, communicate risks and solutions, work equitably and use our platforms to elevate other voices, such as Indigenous and historically marginalized peoples (e.g. Peppoloni and Di Capua, 2017; Nwankwoala, 2019; Stein et al., 2024).

It's a challenging task to make equitable and climate justice projects, and one of the reasons is the variability of the projects, places and groups of people. However, globally, the majority of individuals support climate action (Andre et al., 2024), although perceptions vary on type of carbon removal and between countries (Low et al., 2024). Groups of people can span researchers, collaborators, local communities, representatives that you engage with and people you meet on the way or in the field, companies that work in the area (e.g. mineral exploration or logging), other visitors (e.g. tourists or hunters), to

municipalities and regional districts, and the general public (e.g. Fig. 4). These represent groups of different scale that can be slightly to heavily involved or interested in a project. This is why engaging respectfully, collaborating if applicable, and informing at every stage of a project is crucial.

### 5.4 Lessons learnt and recommendations

Lessons we learned from this work include that to get a project started, it is helpful and necessary to get acceptance from

the relevant parties and rightsholders, the First Nations. It takes time to understand which Nations and which representatives to contact, follow up, and organize meetings and next steps. For this project, we started engagement roughly seven months before fieldwork, and it took a total of ~124-264 hours to engage with Nations within the areas. The engagement work to date took approximately ten months, or roughly a third of the overall project (the first author's PhD project) that also included a feasibility assessment and lab work. Additionally, there is a range of opinions within and between Nations. But

for the most part, the conversations that took place were positive, and there seemed to be an interest in the project. For the Nations that we did not hear from, they might not be interested or do not have the resources to do so. Lastly, early discussions are valuable and helpful to build relationships and understand Nations' priorities. This will be helpful for proposed projects and further engagement later on.

Considerations and recommendations:





1.  It would benefit geoscientists to have relevant insights and training in engagement practices as applicable or to work together with appropriate experts. This could be as part of undergraduate education and with changes in engagement practices, such as departmental or industry guidelines. This would include:

    a.  The potential implications of your planned work and research, especially for local communities and Indigenous Peoples, as applicable. Additionally, effects on focus areas that communities may ask about, even if the project doesn't affect those.

    b.  It is important to bear in mind and reflect on how you might have a different way of knowing or worldview than many communities. This can affect communication strategy, how to approach and the selection of words.

    c.  The current state and history of the area and community. Including any past work by geoscientists, other researchers, and relationships of other groups with the Nation (e.g. Wong et al., 2020; Smith and McPhie, 2022). Also, how the different Nations and alliances work together or their relationships in the area.

2.  During engagement, inform of any background information and be ready to discuss and answer questions. Listen to suggestions and recommendations and present options to collaborate. Incorporate and work with First Nations as much as possible; as the quote says "Nothing about us without us" (FNMPC Conference, 2023).

3.  Follow up during engagement and post fieldwork with updates and disseminate results.

4.  In any of the relevant outputs, credit the help received, e.g. with co-authorships, land acknowledgements in presentations, or acknowledgements in papers. Local place names should be used on maps and in text; and be considerate for other recommendations from Nations (e.g. Wong et al., 2020; Adame, 2021).

5.  To build a successful project, early engagement is the first step towards getting to free, prior and informed consent (e.g. Kennedy and Keenan, 2023; Reid et al., 2024) that is needed, such as within UNDRIP (British Columbia, 2019; Government of Canada, 2024b). Additionally, researchers are responsible to make relevant findings known to relevant parties.

## 6 Conclusions

This paper reviews the steps taken and outcomes of early engagement with multiple First Nations in British Columbia, Canada. The discussions were on informing about a project concept on $CO_2$ storage via mineralization in serpentinite and getting consent for geological fieldwork. We engaged with 21 First Nations or alliances representing 41 Nations or alliances directly or indirectly. The total timelines, hours on the engagement process, representative roles, discussion topics, and depth varied immensely.

The general reception of engagement was positive, and First Nations representatives showed an interest in the project. This resulted in consented geological fieldwork and discussions with multiple Nations on implications, criteria and suggestions.

Throughout the process, we keep learning and reflecting on respectful engagement practices. Additionally, on the roles of geoscientists, especially for $CO_2$ storage implementation. The early engagement and the start of relationship building documented here is the first step for further work for the proposed $CO_2$ storage project. If the proposed project continues,
future work will include more engagement with the Nations and hopefully build toward equitable partnerships. The project's success depends on many technical and socioeconomic considerations, from choosing a site to rock properties in the subsurface, funding, and meaningful and successful community engagement.

**Author contribution**

KS and SP planned the campaign. GMD, RT, and SÓS helped with conceptualization. KS prepared the manuscript with
contributions from all co-authors.

**Competing interests**

The authors declare that they have no conflict of interest.

**Ethical statement**

As discussed in the methods, we submitted an ethics application to the Behavioural Research Ethics Board within the
University of British Columbia (H23-02376). It was decided and agreed upon that for this topic of documenting the engagement process, we did not need to complete the ethics review process. The agreement included keeping Nations and representatives anonymous and not documenting others' knowledge.

**Acknowledgements**

We would like to thank all the people that took time out of their day to discuss with us about the project. As well as the
people who forwarded us to the right contacts and gave us contact information. Additionally, thank you to the cultural monitor who came with us to the field for all the help and discussions. We thank Wendy Bond, Brady Clift, Terre Satterfield, and Ólafur Teitur Guðnason for discussions and insights around this topic.

**Financial support**

This research was funded by a Four Year Doctoral Fellowship and NSERC Canada Graduate Scholarship – Doctoral to KS
and an NSERC Discovery grant to GMD.



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
