# Peer review of "Early engagement with First Nations in British Columbia, Canada: A case study for assessing the feasibility of geological carbon storage"

_EGUsphere, 2024_

## Author Response (AR1)

Replies to reviewers for: Early engagement with First Nations in British Columbia, Canada: A case study for assessing the feasibility of geological carbon storage

**AR 1**

This manuscript has considerable scientific significance to Geoscience Communication and the broader empirical literature given its scope and location. Earning the social license to conduct carbon removal projects within and outside of Canada will require substantial community engagement and involvement. In Canada, project success will require engagement with First Nations and Indigenous communities. However, a dearth of literature currently exists on this specific topic. The manuscript is therefore a meaningful and additive contribution to the literature. This is bolstered by the scientific quality of the manuscript which includes a detailed and transparent accounting of the community engagement, along with a good presentation quality.

**Specific comments**

1. **Does the paper address relevant scientific questions within the scope of GC?** Yes (see above)

2. **Does the paper present novel concepts, ideas, tools, or data?** Yes (see above)

3. **Are the scientific methods and assumptions valid and clearly outlined?** Yes

4. **Are the results sufficient to support the interpretations and conclusions?** Yes

5. **Do the authors give proper credit to related work and clearly indicate their own new/original contribution?** Yes

6. **Does the title clearly reflect the contents of the paper?** Yes

7. **Does the abstract provide a concise and complete summary?** Yes

8. **Is the overall presentation well structured and clear?** Yes

9. **Is the language fluent and precise?** Yes

10. **Are the number and quality of references appropriate?** There are sufficient (quality) references to provide an evidence base that supports the manuscript.

Other:

- Line 12:

  While not assuming this list infers a preferred order of operations, many First Nations and Indigenous communities want to start by building relationships, establishing trust, and making a human connection prior to discussion around consent to proceed with any projects.
  Yes exactly, which is what this study was doing, to clarify we have now changed the wording to reflect this.

- Line 18:

  How to approach the issue of involving the local community in job creation while adhering to potential stipulations from certain governments around the use of union labor in order

to access federal funds from policy initiatives? Granted, this is focused on a project in Canada, but these issues will continue to come up as stipulations to access public funding.

This is an important topic that could be dependent on each grant/fund/initiative. There could be a possibility to put into an agreement at start of a project to prioritize local community for jobs, whether they are part of unions or not. In the case, where union labor is needed a compromise can be done to include others from the local community. Added a sentence on this into section 5.3.1.

- Line 69:

  Did the initial objectives include discussing potential equity partnerships as previously mentioned?
  Not exactly, as this was for the early fieldwork assessing potential prior to getting to that stage. This will be included in the later stages of the proposed project. However, it was discussed with some groups that showed an interest in the project.

- Line 235:

  Is there literature to suggest that any initial outreach should be conducted in partnership with social scientists? Are there any established industry best practices for team composition and expertise related to community outreach pertaining to 'natural' vs. 'social' scientists?
  This has not become best practices yes, as far as we know, but is being discussed (e.g. written in Line 247-255), but not the norm and there is a changing role of geoscientists to social performance (e.g. Mackenzie et al., 2020). Have now added a ref to Mackenzie in Line 242 and to Eberenz in Line 247.

- Line 288:

  "Give yourself time to learn about communities' values, norms and culture and how to communicate and behave respectfully...": This should likely be the first step in any community engagement process.
  -Yes, changed the order here, as this is also shown in Fig. 1.

**AR2**

1. Does the paper address relevant scientific questions within the scope of GC?
   Yes
2. Does the paper present novel concepts, ideas, tools, or data?
   Yes
3. Are the scientific methods and assumptions valid and clearly outlined?
   Yes
4. Are the results sufficient to support the interpretations and conclusions?
   Yes
5. Do the authors give proper credit to related work and clearly indicate their own new/original contribution?
   Yes
6. Does the title clearly reflect the contents of the paper?
   Yes

7. Does the abstract provide a concise and complete summary?
   Yes
8. Is the overall presentation well structured and clear?
   Yes
9. Is the language fluent and precise?
   Yes
10. Are the number and quality of references appropriate?
    Yes

Overall, this paper raises a critical topic--the importance of consultation with Indigenous communities within geoscientific research. Too often, it is argued that a lower level of ethics review and engagement is necessary because many geoscientific projects do not explicitly engage with people; however, as the authors note, this does not recognize longstanding Indigenous rights to consultation and cultural connections to land and landscape. This piece will significantly impact geoscientific work done in Canada and elsewhere.

I think one area of revision, or more accurately, addition, could be in the recommendations section. For example, the authors mention that the UBC ethics review board deemed the research as not warranting a full review, and I found myself curious about the authors' response to this--does this align with their consultation model? Additionally, it would be good to see the recommendations section fleshed out with more actionable goals for institutions to follow to help support the model the authors recommend,

 - Added a new item (new nr. 5) into the recommendations in Line 385-387.

**R3 - Giacomo Medici**

General comments
Novel paper in the field of the energy transition. Please, follow my suggestions to bring the impact out and enlarge the views.

Specific comments
Line 23. Provide detail on Carbfix technology due to the fact that you mention it multiple times. – In line 23 added 'via mineralization' and there is more information in section 2, but also added a bit more about Carbfix there.

Lines 21-26. Mention the possibility to combine $CO_2$ storage with geothermal energy production to reach climate goals on reduction of $CO_2$. Please, refer to those papers that are relevant to $CO_2$ storage / geothermal energy in basaltic and sedimentary rocks:
- Buscheck, T. A., Bielicki, J. M., Edmunds, T. A., Hao, Y., Sun, Y., Randolph, J. B., & Saar, M. O. (2016). Multifluid geo-energy systems: Using geologic $CO_2$ storage for geothermal energy production and grid-scale energy storage in sedimentary basins. Geosphere, 12(3), 678-696.
- Medici, G., Ling, F., Shang, J. (2023). Review of discrete fracture network characterization for geothermal energy extraction. Frontiers in Earth Science, 11, 1328397.
- Added a sentence in line 26 with the citations.

Line 51. Please, specify the 3 to 4 specific objectives of your research by using numbers (e.g., i, ii and iii).

- The reason we did not do that was because that could suggest a priority order which we wanted to not be assumed (as all are as important)

Lines 156-214. Large part of the discussion with no references. The purpose of the discussion is to merge the new insights of the manuscript with previous results and ideas in your field.

- Added a few references in section 5.1 and 5.2

Also: have added new references in
- Line 34, 270 & 349 (Bushman, 2024)
- Line 31 & 245 (Eberenz et al., 2024)
- Line 309 (Hunt, 2013)

Lines 412-607. Please, integrate the relevant literature suggested above on the importance of mafic and sedimentary rocks on the energy transition.
- Added

Figures and tables

Figure 3. There is room to make the figure larger.  - Changed

Figure 3. Increase the font size of the words. – Changed

Figure 4. Better tectonic setting if you refer to the trap system for the carbon dioxide. We're at a too large scale with "geodynamics". Just think about the right terminology. – Changed to 'tectonic setting'

Figure 4. There is a link with geothermics, you should mention it also in the introduction. – Have added 'geothermics' into fig, but not into intro as we don't think it's a bigger factor than the other ones (which are not listed).

**Editor – Louise Arnal**

Dear Authors,

Based on the reviewers' comments and my own evaluation of your manuscript, I would like to request that you undertake major revisions. While the manuscript highlights an interesting engagement activity, I believe it would benefit from a more in-depth and robust qualitative analysis. Strengthening this aspect would significantly enhance its contribution as a research article in Geoscience Communication. Here are my main recommendations for improvements:

- Context & hypothesis: The manuscript should provide more thorough context and formulate a clear hypothesis and/or manuscript objectives (besides presenting the approach used). For instance, it would be helpful to know what the current practices are for engaging local communities in similar projects (at least within BC or Canada) and how the proposed model differs from those practices or addresses specific challenges. This could serve as the foundation for a hypothesis.

- Added a paragraph into section 2 about current practices and a qualitative hypothesis for this study. We are studying: 1) test and model an engagement process (fig 1); 2) timelines and depth, 3) perspectives that impact fieldwork/carbon storage?

- Discussion: It would be useful to hear about where the presented model has been successful (mostly addressed it would be useful to link these successful aspects to the challenges identified earlier - see point above), where it has fallen short (currently under-explored), and how you propose to revise the model in the future (still lacking).

We have added a new section so it is clear (Section 5.5. Summary and evaluation of objectives). Some of the text was moved from 5.4 to this new 5.5.

Additionally, the text has more discussion on the hypothesis and objectives in these sections (numbers from comment above):

1)      Section 4 & 5.1: General about what happened and if/when the engagement system worked and Fig. 2 (=where the engagement system was successful and not successful).
2)      Section 4 & 5.1: Timelines and depth outlined and discussed and Fig. 3
3)      Section 5.1 & 5.2: Perspectives, conversations discussed and in Section 5.3 a discussion of general what this means for responsibilities.
- Where the engagement did not work (no engagement managed) and where we weren't allowed to sample: currently written in 1st paragraph in 5.1 and 4th paragraph in 5.2.
- Revise the model: It will depend on each scenario and location, and thus, recommendations are relatively ambiguous. Section 5.4 discusses this, but I have now added nr 5. Also changed 5.4 to be about general communities instead of Nations. And in 2nd paragraph in Section 5.3. there is mentioned the guidelines document we are making for our earth department as an example that other groups can do.

- Generalizability of the methods and findings: The manuscript would benefit from a discussion of the extent to which the results are applicable to other geographical or research contexts versus how localized the model and outcomes are.

- The considerations section in 5.4 has been changed to be about general local communities (rather than First Nations).

-Added a new consideration (nr. 5) as a general recommendation for departments and companies up to the federal scale.
- Minor aspects: There are repetitive elements in the discussion section that could be removed to make the manuscript more concise.

- Sentence taken out in line 178 (as repeat from outcome section).

- Sentence taken out in line 194.

- Phrase taken out in line 226.

- Phrase taken out in line 323.

- Phrases taken out in lines 356-361.

Please address these recommendations, as well as all reviewer comments you received so far, in your revised manuscript.

Thank you for engaging in the GC review process.

Louise Arnal